# Copper Wire Bonding: A Review

**DOI:** 10.3390/mi14081612

**Published:** 2023-08-16

**Authors:** Hongliang Zhou, Andong Chang, Junling Fan, Jun Cao, Bin An, Jie Xia, Jingguang Yao, Xiaobin Cui, Yingchong Zhang

**Affiliations:** 1School of Mechanical and Power Engineering, Henan Polytechnic University, Jiaozuo 454000, China; cadzmy@163.com (A.C.); a1951198730@163.com (B.A.); x6868mrtx@163.com (J.X.); yaojingguang1205@163.com (J.Y.); xiaobincui@hpu.edu.cn (X.C.); 2School of Chemical and Environmental Engineering, Jiaozuo University, Jiaozuo 454000, China; jzufanjunling@163.com; 3Nanjing High Speed Gear Manufacturing Co., Ltd., Nanjing 211100, China; 212005010007@home.hpu.edu.cn

**Keywords:** Cu wire bonding, FAB morphology, bonding reliability, intermetallic compounds, applications of simulations

## Abstract

This paper provides a comprehensive review on copper (Cu) wire bonding. Firstly, it introduces the common types of Cu wire available in the market, including bare Cu wire, coated Cu wire, insulated Cu wire, and alloyed Cu wire. For each type, their characteristics and application areas are discussed. Additionally, we provide detailed insights into the impact of Free Air Ball (FAB) morphology on bonding reliability, including its effect on bond strength and formation mechanisms. Next, the reliability of Cu wire bonding is analyzed, with a focus on the impact of intermetallic compounds and corrosion on bonding reliability. Specifically, the formation, growth, and stability of intermetallic compounds at bonding interfaces are discussed, and their effects on bonding strength and reliability are evaluated. The detrimental mechanisms of corrosion on Cu wire bonding and corrosion inhibition methods are also analyzed. Subsequently, the applications of simulation in Cu wire bonding are presented, including finite element analysis and molecular dynamics simulations, which provide important tools for a deeper understanding of the bonding process and failure mechanisms. Finally, the current development status of Cu wire bonding is summarized, and future research directions are discussed.

## 1. Introduction

Wire bonding is widely recognized as a prominent interconnection technique within the realm of microelectronic packaging due to its inherent flexibility and ease of implementation. In this domain, Au wire has traditionally been favored as the primary choice for bonding wire material, owing to its exceptional ductility, corrosion resistance, and oxidation properties. Nonetheless, the substantial surge in gold prices witnessed in recent years has spurred the gradual emergence of Cu, Al, and Ag wire as more economically viable alternatives [1]. Moreover, bonding wire composed predominantly of Cu or Ag, boasting heightened thermal conductivity and adequate electrical conductivity, effectively caters to the semiconductor industry’s mounting demand for diminutive form factors and ultra-fine pitch packaging, thereby fulfilling the heightened input/output (I/O) requirements. In contrast to gold wire, the increased rigidity exhibited by Cu wire renders it better suited for fine-pitch bonding applications. Furthermore, the kinetics of intermetallic compound (IMC) formation in Cu-Al bonding proceeds at a more leisurely pace than those in Au-Al bonding, thereby mitigating the formation of deleterious IMCs [2]. Consequently, there is a burgeoning interest in transitioning from Au wire to Cu, Al, and Ag wire within the purview of wire bonding technology. Table 1 shows the key performance of several common bonding wires. Nevertheless, it is worth noting that pure Ag is afflicted with concerns pertaining to Ag ion migration and corrosion, despite its cost effectiveness in relation to Au wire but still being significantly higher than Cu wire. Conversely, Al wire confronts hurdles associated with subpar mechanical performance, susceptibility to oxidation, and erratic behavior, thereby hampering its widespread adoption within the microelectronic packaging landscape. Consequently, the field of Cu wire bonding technology has emerged as a focal point of heightened scrutiny and comprehensive investigation [3,4].

Cu wire offers several advantages in wire bonding [5,6]:(1)High electrical and thermal conductivity.(2)It maintains high strength even at elevated levels of elongation.(3)Good arc stability.(4)Reduced formation of intermetallic compounds during the bonding process.(5)Formation of a favorable ball shape during bonding when protected by inert gas.

However, Cu wire also has its drawbacks. It is highly susceptible to oxidation in ambient air, and its higher hardness compared to other wire materials makes it prone to damaging the bonding pad during the bonding process [7]. The increased bond strength of Cu wire makes it unsuitable for bonding with low-hardness bonding pad materials [8]. To address the surface oxidation of Cu wire during Cu wire bonding, ultrasonic bonding has been introduced to prevent surface oxidation of the first bond (FAB) by introducing an oxide film of higher hardness than Cu. Researchers have also started investigating Ni-based high-hardness metal bonding pads to address the issue of damage to Al pads [9]. Ni has a hardness approximately 50% higher than Cu and four times higher than aluminum, making it capable of withstanding higher stresses generated by Cu ball bonding and preventing damage during impact, thereby potentially replacing Al-bonding pads in specific applications. One of the reasons why Cu wire has not completely replaced gold wire is because Cu has a narrower process window compared to gold. Having a large process window is crucial for stable Cu processing, as it ensures optimal quality production, process portability, and stability [10]. Studies have shown that the hardness of Cu wire decreases with increasing purity levels [11]. Hence, using high-purity Cu wire in the bonding process can alleviate damage to the bonding pad caused by Cu wire. However, high-purity Cu wire incurs higher manufacturing costs, making it unsuitable for large-scale production. The study conducted by Gurbinder Singh et al. [12] successfully achieved reliable wire bonding using lower-purity Cu wire. B. Czerny et al. [13] investigated the fatigue behavior, bonding parameters, and aging conditions of different materials used for wire bonding, including Al, AlMg, Cu, and Al-coated Cu (CucorAl). The results showed that Cu wire bonding exhibited significantly better fatigue performance and static shear strength compared to the other materials. The next best performance was observed for AlMg and Al wire bonding. This research highlights the potential of low-purity Cu wire materials in wire bonding and their ability to contribute to cost reduction in Cu wire bonding processes.

## 2. Types of Cu-Bonding Wire

In recent years, Cu-bonding wire has been widely utilized in microelectronics packaging. Currently, there are various types of Cu wire available in the market, including bare Cu wire, coated Cu wire, insulated Cu wire, and Cu alloy wire.

### 2.1. Bare Cu Wire

As one of the alternatives to Au, bare Cu wire has been extensively studied as a Cu-bonding wire. However, bare Cu wire faces challenges such as high hardness, oxidation susceptibility, and bonding failure. Due to the lack of an insulation layer, bare Cu wire may not be suitable for certain packaging applications and requires additional insulation measures. Consequently, the development of bare Cu wire has been significantly limited. When comparing the reliability of bare Cu wire and Palladium-coated Cu (PCC) wire, Tomohiro Uno et al. [14] found that the failure time of bare Cu wire was shorter than that of PCC wire, indicating that bare Cu wire was not as reliable in bonding as PCC wire, and there was also serious corrosion at the bonding interface of bare Cu wire.

### 2.2. Coated Cu Wire

Coating the surface of Cu wire with a layer of highly antioxidative or corrosion-resistant metal can enhance its resistance to oxidation, ensure a visually appealing metallic luster, improve bonding performance, and significantly extend the shelf life of bonded Cu wire products after unpacking. Currently, the most extensively studied types of metal coatings are noble metal coatings (such as Au, Ag, Pt, and Pd) and corrosion-resistant metal coatings (such as Ni, Co, Cr, and Ti). Generally, introducing elements like Au, Ag, Pd, and Ni into Cu wire can enhance the reliability of Cu wire bonding in microelectronics packaging. However, except for Pd, most elements tend to cause shape deformations of the FAB, such as spearhead or non-spherical shapes [15]. Nonetheless, some scholars have discovered that when the thickness of the Pd coating on Cu wire is relatively thin, it can also result in FAB eccentric ball [16]. Mitsubishi Electric Corporation [17] has invented a Ni-coated Cu wire that exhibits a certain level of reliability in experiments. A comparison was made between Pt, Pd, and Ni coatings on Cu wire, and the results revealed that PCC wire exhibited the best performance, with stronger oxidation resistance than Ni, better processability than Pt, higher mechanical strength than pure Cu-bonding wire, moderate hardness, and excellent bonding ball formation [18,19]. In terms of electrical characteristics, PCC wire and bare Cu wire have similar electrical resistances, and their melting currents are almost the same [20,21]. Table 2 shows the basic performance comparison between PCC wire and bare Cu wire. Moreover, the research indicated that Pd does not participate in the formation of Cu-Al IMCs [22]. In the study conducted by Adeline B.Y. Lim et al. [23] on the Pd-Cu wire bonding with Pd at the bonding interface, they observed the presence of voids in the Pd-Cu wire bonding IMC layer. However, these voids did not affect the tensile strength of the ball bond. In contrast, in the Pd-Cu wire bonding without Pd at the bonding interface, no nanovoids were found. This indicates a close correlation between the presence of Pd and the formation of nanovoids. As shown in Figure 1. Hence, PCC wire holds broad prospects for applications in Cu wire materials [24]. Consequently, extensive research has been conducted on PCC wire. Typically, PCC wire is produced by electroplating palladium onto the drawn Cu wire. However, this method poses significant environmental harm. Although various methods have been developed to reduce the hazards associated with electroplating plating, they still have negative environmental implications [25,26,27,28]. Therefore, scholars have explored more environmentally friendly approaches to produce Pd-coated Cu wire. For example, Cao Jun et al. [29] utilized a halogen-free direct palladium coating process to produce uniformly PCC wire, significantly reducing the environmental impact of palladium electroplating. Furthermore, the bonding process of PCC wire only requires the use of nitrogen gas [30,31], whereas a combination of nitrogen and hydrogen gas is preferred for bare Cu wire bonding. Currently, research is ongoing to address the aforementioned issues, and it is evident that the study of coated Cu wire holds immense significance in the realm of Cu-bonding wire.

### 2.3. Insulated Cu Wire

Insulated Cu wire is composed of a Cu wire core and an insulation coating. This insulation coating serves the purpose of preventing short circuits and oxidation issues in Cu wire, making insulated Cu wire technology a potential solution for fine and ultra-fine-pitch wire-bonding applications [32]. Due to only adding a layer of insulation coating on the Cu wire, the conductivity of insulated Cu wire is close to that of bare Cu wire. The use of insulated wire for wire bonding was developed over two decades ago. As microelectronic packaging becomes increasingly complex, there will continue to be a certain market demand for wire bonding using insulated wire in the future. Insulated wire also possesses the characteristic of preventing Cu wire oxidation in the air. For example, Hyun-Maria Lykova et al. [33,34,35] designed a temporary protective coating using a Cu-Cu organic self-assembled monolayer (SAM) to inhibit Cu oxidation. Research results showed that the SAM coating could prevent Cu from oxidizing for at least 12 h in the air. Additionally, Cu wire bonding with this protective layer exhibited better quality. When comparing the bonding of insulated Cu wire and bare Cu wire, HungYang Leong et al. [36] found that samples with insulated Cu wire consistently produced larger first bond sizes, indicating that less energy was required for FAB formation in insulated Cu wire, as shown in Figure 2. Furthermore, no organic insulation residues were observed on the FAB surface, suggesting that the organic insulation layer was completely burned during FAB formation. Insulated Cu wire demonstrated better performance in terms of ball shear strength, wire tensile strength, and IMC coverage compared to bare Cu wire. However, the stitch bond strength of insulated Cu wire was 24% lower than that of bare Cu wire. Therefore, further process optimization is needed to enhance the bonding strength of insulated Cu wire. Moreover, the high processing cost of insulated Cu wire, the risk of insulation layer delamination, and the lack of sufficient market demand contribute to its low market share.

### 2.4. Cu Alloy Wire

Alloy Cu wire is a material composed of Cu and other alloying elements, widely used in electronic packaging and microelectronics for bonding applications. Compared to traditional bare Cu wire, alloy Cu wire exhibits superior performance and reliability. By adjusting the composition of the alloy, the mechanical properties, corrosion resistance, and conductivity of the Cu wire can be optimized. Currently, research on Cu alloy wire is still relatively limited. However, there is significant potential in adding certain precious metals or rare earth elements to Cu. The addition of elements such as Mg, Ru, Nb, Rh, Y, Cd, and Zn to Cu wire can improve their oxidation resistance, whereas the inclusion of Zr, Ti, and Fe can refine grain structure and reduce the hardness of Cu wire, However, most of the added elements will reduce the conductivity of Cu wires, such as Be, Mg, Al, etc. [37,38,39]. The addition of precious metal elements is beneficial for improving ball bonding shape, enhancing bond strength, and suppressing surface oxidation. Jun-Ren Zhao et al. [40,41] studied the performance of Cu wires by adding trace amounts of Au, Pd, and Pt individually. The results showed that the addition of Au or Pd had a grain refining effect, whereas the addition of Pt significantly increased the resistance of Cu wire. However, Cu wire with added Pt exhibited the best electrical fatigue life. Among them, Cu wire with the addition of 0.30–0.39 wt% Pt, 0.20–0.29 wt% Au, and 0.10–0.19 wt% Pd demonstrated optimal overall performance. Moreover, micro-alloyed wires exhibit higher electrical fatigue strength compared to PCC wires. Shen-Teng Hsu et al. [42] compared the reliability of PCC wire with a Cu alloy wire that was additionally alloyed with Pt, Au, and Pd. The results revealed that the Cu alloy wire with added noble metals exhibited better reliability compared to PCC wire. However, an excessively high Pt content in the alloyed Cu wire can lead to a rapid increase in electrical resistance. Motoki Eto et al. [43] conducted experiments to understand the effects of added elements, Pd and Pt, in PCC wire cores and their corresponding impacts on corrosion resistance under high-temperature life storage (HTSL). The results revealed that the addition of Pd and Pt in PCC wire led to a decrease in bond reliability under HTSL, especially in the pin-jointing area. PCC wire without added metals exhibited degradation of the Cu core material, possibly due to the presence of voids near the pins. This indicates that it is important to control the content properly when adding other elements. The appropriate proportion can enhance the performance of Cu wire, whereas an imbalance may result in a decrease in bond reliability.

In conclusion, controlling the content of various alloying elements within certain proportions can prevent oxidation, refine grain structure, or slow down IMC growth. However, further in-depth research is still necessary in order to produce alloy wire that meets the required performance criteria in all aspects.

## 3. FAB Morphology

FAB refers to the formation of a free spherical structure of air during the metal bonding process. FAB has a significant impact on the reliability of the bond, with the most prominent influence being on bond strength. A uniform and stable FAB facilitates the formation of a robust solder joint, leading to enhanced bond reliability. Researchers such as Hong Meng Ho et al. [44] have studied the ball formation properties of Cu wire and found that a larger Electronic Flame Off (EFO) current or shorter ball formation time results in better control of FAB diameter standard deviation. The study also revealed a correlation between the hardness of Cu solder balls and EFO parameters, indicating that higher EFO currents lead to a softer FAB. For Cu wire bonding, to obtain a softer FAB and minimize stress during the initial bonding impact, it is advisable to form a FAB within a shorter current-on time. A.B.Y. Lim et al. [45] compared the formation of a FAB and differences in wire bonding between PCC and bare Cu wire, as shown in Figure 3. It was observed that under varying gas types and EFO currents, PCC wire exhibited the formation of irregular solder balls. The distribution of Pd on the FAB was also influenced by the type of protective gas and EFO current, with the EFO current being the primary factor. Adjusting the EFO current resulted in different Pd distributions. The study further observed gaps in the bond interface after PCC wire bonding, and these gaps were closely related to the presence of Pd, as depicted in the figure. After annealing PCC bonds for 24 and 168 h at 175 °C, two layers of IMCs were observed. The experimental findings had significant implications for understanding the mechanism of gap formation at the bond interface due to Pd distribution and the impact of a FAB on wire bonding reliability.

Du Yahong et al. [46,47] conducted wire bonding experiments using PCC wire on Al pads, establishing a model for the distribution of Pd on the FAB during wire bonding. They also investigated the effects of Pd on the Cu-Al interface reaction during ultrasonic welding. The results showed that a higher EFO current resulted in a smaller region of Pd distribution on the FAB surface for the same FAB diameter-to-wire diameter ratio. With an increase in current-on time, the coated Pd elements could dissolve into the Cu matrix, forming a PCC alloy on the entire surface of the FAB, which acted as a barrier to prevent oxidation of the Cu wire. The study further found that the Pd coating could increase the ball shear strength of bare Cu wire bonding by over 50%. However, FAB with the highest Pd coverage exhibited the lowest ball shear strength during bonding. This may be attributed to excessive Pd elements leading to the formation of gaps and cracks at the bond interface. These gaps and cracks have a significant impact on bond reliability during shear testing, indicating that controlling the distribution of Pd on the FAB is crucial for achieving high-quality bond interfaces.

## 4. Bonding Reliability

Bonding reliability encompasses the physical and chemical characteristics of solder joints or bonding interfaces formed during the bonding process, as well as their performance and stability under actual operating conditions. The performance and lifespan of electronic devices are significantly influenced by bonding reliability. In certain environments, wire bonding may lead to the formation of detrimental IMCs, cracks, or corrosion at the bond interface, consequently resulting in bond failure. The strength of solder joints or bonding interfaces is a critical factor determining bonding reliability, as insufficient strength can lead to solder joint fracture or bond interface failure. In microelectronic devices, shear testing of wire bonds is crucial for evaluating the strength of bond solder joints. However, the actual mechanisms involved in shearing Cu bonded to an Al-bonding pad are not yet fully understood. To uncover the detailed mechanisms of shear testing, Subramani Manoharan et al. [48] conducted shear tests on Cu-Al wire bonds. The research findings revealed that complex stress behaviors occur within the wire bond interface during the application of shear stress, contributing to overall compression and tensile states, consequently resulting in simple or complex failure modes. The results indicate that the shear force initially increases based on the remaining thickness of Al underneath the bond, reaches a peak, and eventually decreases due to the growth of IMC and variations in the shearing mode. Compared to noble metals, Cu possesses higher hardness, necessitating greater bonding force and ultrasonic energy for bonding to the bonding pad. The higher bonding force leads to the occurrence of Al splashing on the bonding pad, which may cause pad delamination and bond failure. Gu Liqun et al. [49] compared Cu-Al and Au-Al wire bonding and observed the presence of cracks at the Cu-Al bond interface, resulting in leakage issues during electrical experiments. However, in comparison to Au wire bonding processes, the Cu wire bonding process exhibits slower thermal diffusion rates and lower electromigration (EM) flux. The reduction in EM flux in Cu wire bonding significantly minimizes crack formation, thereby greatly extending the lifespan of Cu wire bonding processes [50].

### 4.1. Influence of IMCs on the Reliability of Cu Wire Bonding

The reliability of the wire bond interface in electronic packaging primarily depends on the microstructure and thickness of IMCs formed between the chip pad and the bonding interface [51]. Extensive research has been conducted by scholars on the influence of IMCs on the reliability of Cu wire bonding. Yahong Du et al. [52] systematically studied and compared the reliability and failure analysis of Cu wire and Au wire bonding processes on Al substrates after successful bonding. The results revealed that the Au wire bond interface exhibited thicker IMCs, as well as more voids and cracks, compared to the Cu wire bond interface, indicating a close correlation between the growth of IMCs and the presence of cracks.

Pd and other metals play a crucial role as controlling components in the reliability of Cu-Al wire bonding. It has been observed by researchers that Pd does not participate in the formation of Cu-Al IMCs. Tomohiro Uno et al. [14] found that PCC exhibited strong reliability in high-humidity environments while studying the bonding reliability of PCC. The presence of Pd in PCC improves bonding reliability by controlling diffusion at the bonding interface and IMC formation. Studies by Wentao Qin et al. [53] investigated the mechanism of Pd coating in wire bonding, and it was found that the dominant IMC formed when using bare Cu wire for wire bonding was Cu_3_Al_2_ after unbiased high-acceleration stress testing (uHAST), whereas (CuPd_x_)Al was predominantly formed under PCC wire bonding. The HAADF image of the PCC wire bonding interface is shown in Figure 4. This indicates that Pd coating inhibits the formation of (CuPd_x_)_3_Al_2_. This inhibition is attributed to the improved thermodynamic stability of (CuPd_x_)Al, which reduces the kinetic driving force for the phase transition to (CuPdx)_3_Al_2_. The cathode/anode area ratio of (CuPd_x_)Al is lower than that of Cu_3_Al_2_, resulting in a lower corrosion rate of (CuPd_x_)Al and improved reliability of wire bonding. A comparison has also been made between the reliability of PCC wire bonding and Au wire bonding, revealing longer failure times for PCC wire bonding and slower growth of IMCs. Kuan-Jen Chen et al. [54] conducted extreme sulfurization tests to compare the corrosion resistance of PAC (Palladium-coated Cu wire with a flash-gold layer, which combines the characteristics of PCC and gold-plated Cu wire) and PCC. Both types of Cu wire were wire bonded to Al pads, and it was found that the gold-plated layer of the PAC wire effectively enhanced its resistance to sulfurization corrosion. The presence of the gold-plated layer resulted in different levels of corrosion on the wedge bonds for the two wire types. The gold-plated layer of the PAC wire strengthened the bond and maintained low resistivity characteristics after sulfurization testing, suggesting that the presence of precious metal elements may contribute to the bonding performance of Cu wire. It has been discovered that different molding compounds also have an impact on the growth rate of IMCs [55].

Various phases of Cu-Al IMCs have been identified as CuAl_2_, CuAl, Cu_4_Al_3_, Cu_3_Al_2_, Cu_9_Al_4_, and Cu_3_Al [56,57,58,59]. Through experiments, KIM H G et al. [60] found that Cu-Al IMCs form in the order of CuAl_2_, CuAl, and Cu_9_Al_4_ in Cu-Al wire bonding, whereas Cu-Au IMCs form in the order of (Au, Cu), Cu_3_Au, and (Cu, Au). The atomic diffusion between Cu and Au and the volume difference of IMCs result in the formation of voids near alumina fragments. Cu-Al IMCs’ TEM image is shown in Figure 5. Through the analysis of the complete IMC at the bonding interface, Chien-Pan Liu et al. [61] discovered that the ball bond interface between the Cu wire and Al-bonding pad consists of layered structures composed of Cu, Cu_9_Al_4_, CuAl_2_, and Al. They also found that bonding failures frequently occurred between the Cu_9_Al_4_ and CuAl_2_ layers due to high current corrosion rates and the presence of high-concentration interfacial voids, which aligned with the previous findings on IMCs. They further developed a predictive model for IMC growth in Cu wire processing technology, which could help reduce Cu-Al interface failures in practical applications. T. Joseph Sahaya Anand et al. [62] bonded Cu wire to interfaces of Al, AlSiCu, and NiPdAu pads, and it was observed that the growth rate of IMCs followed the sequence Al > AlSiCu > NiPdAu. They also found that a faster IMC growth rate led to accelerated crack formation. Hyun-Woong Park et al. [63] studied the effects of Pd and Pd-Au coatings on Cu wire bonding. The results showed that noble metal coatings formed a fully solid solution at the Cu-Al bonding interface, potentially impeding the diffusion of Cu-Al IMCs, especially at the Cu_9_Al_4_ IMC, which may delay the IMC growth rate [64]. The formation of a fully solid solution is believed to enhance the bonding reliability by protecting the IMCs from crack propagation. Adeline B. Y. Lim et al. [65] found that CuAl exhibits the highest hardness and stiffness among Cu-Al IMCs. Cu_9_Al_4_ is the most susceptible to corrosion among IMCs in Cu wire bonding, whereas CuAl_2_Cu shows better corrosion resistance [66]. One of the failure factors in wire bonding components is the corrosion of the intermetallic layer [67,68].

### 4.2. Influence of Corrosion on the Reliability of Cu Wire Bonding

After wire bonding, the bonding interface is encapsulated with epoxy molding compound. However, epoxy molding compound can absorb moisture from the air. The absorbed moisture can dissolve ionic substances in the molding compound and form a thin electrolyte layer around the metal materials. Due to the different electrochemical activities of Cu-Al IMCs, Cu, and Al, galvanic corrosion may occur at the Cu-Al bonding interface [69]. As cracks caused by corrosion propagate towards the center of the bonding interface, crevice corrosion may also occur within the narrow region of the crack, further accelerating crack propagation. Ultimately, the reliability of Cu wire bonding is compromised. Figure 6 shows the situation at the beginning of corrosion.

Corrosion can have several impacts on bond reliability [67,70,71,72]:(1)Formation and stability of bond interface: corrosion may lead to the formation of IMCs at the bond interface, affecting the stability of the bond. Some IMCs formed during corrosion may enhance bond strength and reliability, whereas others can cause interface embrittlement and delamination, thereby reducing bond reliability;(2)Material loss and weakening: corrosion can cause material loss and weakening, resulting in thinning of the bond area or the formation of voids, which can impact bond strength and reliability;(3)Electrical and thermal properties of the interface: corrosion can alter the electrical and thermal properties of the bond interface, affecting the electrical and thermal performance of the device and, in turn, its reliability;(4)Accumulation of corrosion products: the accumulation of corrosion products can lead to local stress concentration in the bond area, increasing the risk of bond failure.

Structural stability of the interface: corrosion may destabilize the structure of the bond interface, affecting the long-term reliability of the bond.

Therefore, for applications involving Cu wire bonding and other electronic devices, it is crucial to conduct in-depth research on corrosion and implement effective corrosion inhibition measures to ensure bond reliability and stability.

#### 4.2.1. Effect of Halogen Ion on Corrosion of Cu Wire

Studying the effect of halogen ions on Cu wire corrosion has potential help in improving bonding reliability. Studies conducted by Omid Mokhtari et al. [73] investigated the corrosion behavior of Cu-Al IMC layers in NaCl solution. The comparison before and after corrosion is shown in Figure 7. The results showed that only CuAl IMC was corroded among Cu_9_Al_4_, CuAl, and CuAl_2_ IMCs. The largest potential difference was observed between Cu_9_Al_4_ and CuAl, indicating the formation of a galvanic cell reaction at the interface between Cu_9_Al_4_ and CuAl. This implies that CuAl acts as the anode, which may be the cause of corrosion in the intermetallic layer. Chien-Pan Liu et al. [74] conducted similar studies on Au, Cu, PCC, and Ag alloy wire. The influence of chloride ion concentration on bonding reliability showed similarities to the aforementioned experiments. The broken Cu wire with chloride distribution is shown in Figure 8. The measured corrosion current density (i_corr_) decreased in the order of Cu wire > PCC wire > Ag alloy wire > Au wire, and the open circuit potential (E_oc_) decreased in the order of Au wire > Ag alloy wire > PCC wire > Cu wire. With an increase in chloride ion concentration or a decrease in pH value, E_oc_ shifted towards smaller values, and i_corr_ sharply increased in each metal wire sample. Cheng-Fu Yu et al. [75] discovered during their investigation of the failure mechanism of Cu wire bonding that initial cracks were formed through the ultrasonic extrusion effect during the thermosonic bonding process. With increased aging time, cracks propagated towards the center of the ball bond, and chloride ions diffused through the cracks to the center of the ball bond. This diffusion led to corrosion reactions between chloride ions and Cu-Al intermetallics, resulting in bond failure of the Cu wire. Adeline B.Y. Lim et al. [76] found in their study on Cu, Al, and Cu-Al IMCs that the corrosion current of IMCs in a 25 ppm chloride solution at pH 6 increased with an increase in Al content. Nick Ross et al. [77] made new findings regarding this corrosion behavior: (1) the cathodic half-cycle of corrosion was initially supported by the reduction in dissolved oxygen and later taken over by hydrogen evolution during the late stages of acute Al pad corrosion. (2) Hydrogen evolution was identified as part of the cathodic reaction causing Cu wire separation during the corrosion of Al pads under acidic chloride conditions. (3) Peripheral Cu/Al bimetallic contact drove hydrogen evolution and significantly accelerated the corrosion process of the Al pads, leading to bond separation failure.

#### 4.2.2. Measures to Reduce and Prevent Interface Corrosion

To mitigate corrosion issues, two common approaches are often employed: the use of metal coatings on Cu wire and the use of molding compounds with low chloride concentrations. However, the exact mechanisms behind the effectiveness of these methods are not fully understood. Researchers have focused on several aspects, such as the role of halide ions, as the primary cause of Cu-Al corrosion. Yuelin Wu et al. [78] studied the role of Pd and chloride concentration. The findings indicated that the addition of Pd and reducing chloride concentration decreased the galvanic corrosion rate between Cu and Cu_9_Al_4_. Furthermore, in electrolytes with low chloride concentration, the galvanic effect was completely eliminated with a significant amount of Pd addition (9 wt%). By reducing chloride concentration, the galvanic corrosion rate between Cu_9_Al_4_ and CuAl_2_ was reduced due to the lower anodic dissolution rate of CuAl_2_. However, due to the higher cathodic activity of Cu_9_Al_4_, the addition of Pd increased the galvanic corrosion rate between Cu_9_Al_4_ and CuAl_2_. Nonetheless, it is known that Pd reduces the growth rate of IMCs and related stress accumulation. As a result, the formation of gaps at the Cu_9_Al_4_-CuAl_2_ interface is reduced, leading to an improved bonding success rate. M. Eto et al. [79] also investigated the influence of Pd on corrosion and failure mechanisms in wire bonding. The results showed that the corrosion of Cu-Al IMCs containing Pd was inhibited. The mechanism behind this inhibition was likely due to the formation of a protective passivation film rich in Pd on the IMC surface, which resisted the addition of halide ions and slowed down the corrosion reaction.

Therefore, it is crucial to use molding compounds with low chloride contents and employ metals, as well as improved production processes, to reduce the corrosion rates of IMCs for enhanced bond reliability. In addition, scholars have made many attempts to reduce and prevent corrosion: Motoki Eto et al. [80] employed a novel technique known as post-bonding heat treatment and found that introducing this process effectively inhibited the corrosion progression of Cu-Al IMCs. He also discovered that the corrosion process of Cu-Al pads was influenced not only by the chemical reactions between Cu/Al IMC and impurities from molding compounds but also by mechanical factors such as tensile stress induced by corrosion shapes, products of each Cu-Al IMC, and lattice mismatches.

To eliminate the potential difference between the bi-metals and avoid galvanic reactions, researchers have started exploring the use of Cu-Cu wire bonding instead of Cu-Al bonding. Cu-Cu bonding has shown improved electrical fatigue life compared to Cu-Al bonding and helps mitigate issues related to the vulnerability of Cu bonding to damage the bond pad [40]. Shen-Teng Hsu et al. [42] experimentally confirmed the excellent reliability of Cu-Cu wire bonding. Furthermore, Cu-Cu bonding avoids the impact of the direct growth of Cu-Al IMCs on bonding reliability. K. Abdul Hamid et al. [81] conducted research on the mechanism of electrolyte reactions and chemical substances, comparing the corrosion behavior of Cu-Cu and Cu-Ag systems under different conditions. The results revealed significant morphological changes in the Cu wire bonded on a Ag-plated lead frame (CuAg) in various electrolyte solutions, following the galvanic corrosion model induced by the bi-metallic elements (Cu and Ag) in metallurgical interconnects. Both Cu wire systems experienced severe corrosion when immersed in NaCl. However, Cu-Cu bonding also presents new challenges, such as the higher melting point of Cu (1083 °C) and lower self-diffusion rate. Therefore, further research is needed to gain a deeper understanding of Cu-Cu wire bonding.

## 5. Application of Simulation in Cu Wire Bonding

In Cu wire bonding, simulation offers the following advantages:(1)Cost-effectiveness: by using simulation techniques, virtual testing and optimization can be performed during the design phase, avoiding the costs associated with actual manufacturing and testing. This helps reduce material waste, production cycles, and overall costs;(2)Performance prediction: the performance of wire bonding can be predicted and evaluated in a simulated environment. By simulating the wire bonding process and stress distribution, potential failure points can be identified, and the reliability and durability of the wire can be predicted;(3)Design optimization: simulation tools allow for the optimization of wire bonding designs. By altering parameters such as wire dimensions, geometric shapes, and welding parameters, the impact of different design options on wire performance can be assessed, enabling the selection of the best design to enhance wire bonding performance;(4)Reduction in experiments: simulation techniques can reduce reliance on physical experiments. By simulating different scenarios and operating conditions, various hypotheses and solutions can be quickly tested in a virtual environment, reducing the number of experiments required and saving time.

To further understand the failure modes of Cu wire bonding, researchers have employed finite element analysis to simulate the compression state of the Al-bonding pad during the bonding process. For example, H. C. Hsu et al. [82] studied the parameters for reducing Al pad compression through finite element analysis. The study found that an increase in the contact velocity (C/V) resulted in reduced compression of the Al-bonding pad during the bonding process, and a decrease in the ultrasonic vibration time also reduced the plastic deformation of the Al-bonding pad. The FAB extrusion state during wire bonding is shown in Figure 9. Fa Xing Che et al. [83] investigated the failure modes and mechanisms of Cu-low-k chip wire bonding and bonding pad reliability through wire pull tests and finite element analysis. They conducted wire pull tests by changing the pulling position of the bonded Cu wire, and the results showed that the pulling position influenced the failure force and failure modes. Their finite element analysis model further enhanced the understanding of the failure mechanisms. In order to better predict the wire bonding lifespan, M.D. Hook et al. [84] developed a method to predict the wire bonding lifespan. They analyzed and predicted the bonding of Cu wire and palladium-coated Cu wire on an Al-bonding pad. The results showed that the Cu wire bonding on the Al-bonding pad could be highly reliable at 163 °C without encapsulation or at 155 °C with encapsulation. A. Mazloum-Nejadari et al. [85] established a real-life lifespan prediction model for Cu wire based on the physics of failure approach. Such lifespan prediction methods can be used in the future to predict the quality of various wire bonding packages with different geometries and material combinations within a reasonable timeframe, thus reducing the workload of experiments.

Although wire bonding technology has been widely and extensively used, there is still insufficient targeted research on the mechanisms of micro-bonding seam changes, including formation, deformation, and fracture, mainly due to the challenges of conducting experimental studies. Therefore, researchers have conducted numerous studies on simulating the mechanisms of micro-bonding seam changes.

For example, Beikang Gu et al. [86] investigated the formation and fracture mechanisms of micro-bonding seams during Cu-Cu wire bonding using molecular dynamics simulations. The Cu-Cu wire bonding model is shown in Figure 10. They established a contact model for the nanoscale indentation process between the metal wire and the substrate to simulate the contact process between the Cu wire and Cu substrate. The results showed that micro-bonding seams can form and fracture within an extremely short period of time. At the initial stage of contact between the wire and the bonding pad, attractive forces were generated, leading to bonding. Yangyang Long et al. [87], based on a molecular dynamics model, obtained similar conclusions and found that the formation and fracture of micro-bonding seams occurred continuously during the adhesive process. The presence of pre-existing cracks may assist in the formation of new cracks, and materials with higher stiffness are more prone to micro-crack formation. Due to the increasing depth of research on wire bonding, experimental studies are becoming more time-consuming and challenging. Therefore, the application of simulations in various aspects of wire bonding is expected to become more prevalent.

## 6. Summary and Prospect

Cu wire bonding, as an essential interconnection technology, holds promising prospects in electronic packaging and microelectronics. Through further research and exploration, continuous improvement in the performance and reliability of Cu wire can be achieved, thereby promoting its application in high-performance electronic devices and contributing to the development of the microelectronics industry. Future research in this field can be directed towards the following directions. Firstly, further research and optimization of the material properties of Cu-bonding wire are needed, including compositional control, microstructure manipulation, and surface modification of coated Cu and alloyed Cu wires. By improving the mechanical properties and corrosion resistance of the materials, the bonding strength and reliability of Cu wires can be enhanced. Additionally, studying the morphology of the FAB is an important area of investigation. Exploring the influence of FAB on bonding reliability and developing methods to control FAB morphology can further improve the quality and reliability of Cu wire bonding. Secondly, there is a need to continue investigating the mechanisms of corrosion on Cu wire bonding and to explore more effective corrosion inhibition methods. Last but not least, there should be an accelerated expansion of the application of simulations in the bonding process. Through ongoing research and innovation, Cu-bonding wires are expected to play a more significant role in electronic packaging and microelectronics, contributing to the development of related fields.

## Figures and Tables

**Figure 1 micromachines-14-01612-f001:**
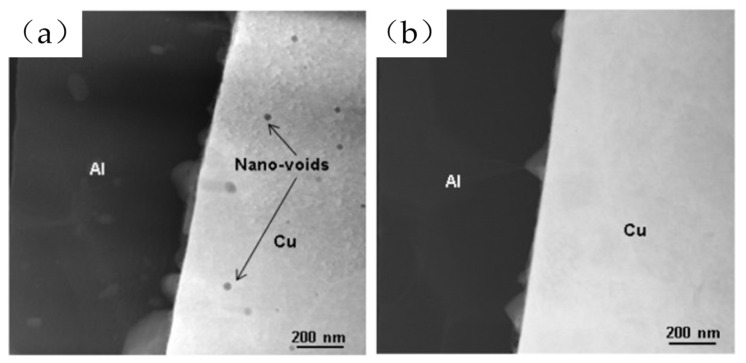
HAADF STEM images of (**a**) the Pd–Cu bond containing Pd showing nano-voids in the Cu (dark spots) and (**b**) the Pd–Cu bond with no Pd [23].

**Figure 2 micromachines-14-01612-f002:**
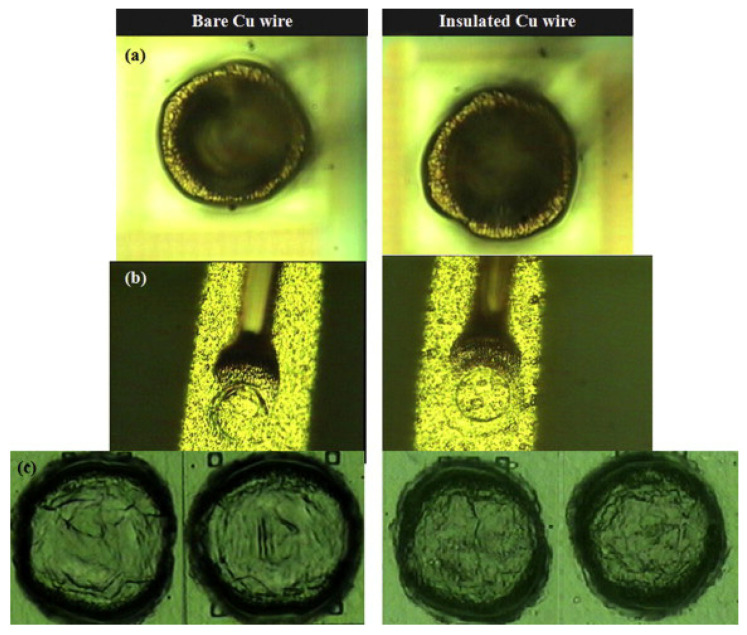
Comparison of bare Cu and insulated Cu wire bonded samples. (**a**) Ball bond, (**b**) stitch bond, and (**c**) IMC coverage and pattern [36].

**Figure 3 micromachines-14-01612-f003:**
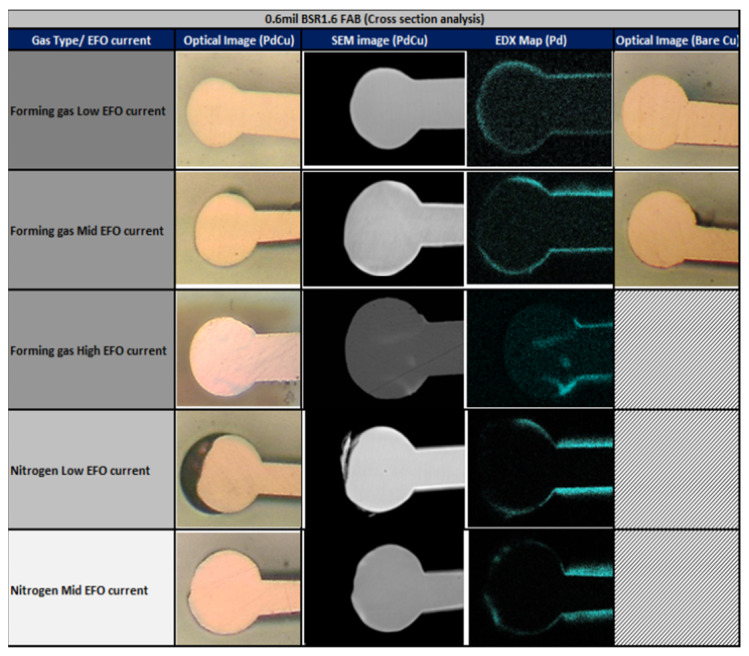
FAB Cross section for PCC and bare Cu wire [45].

**Figure 4 micromachines-14-01612-f004:**
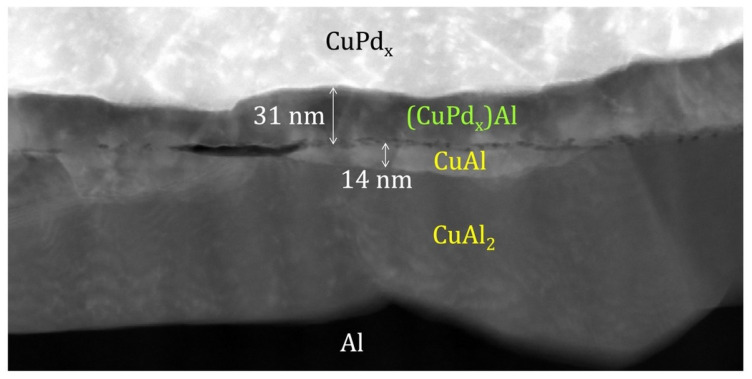
HAADF image of the bonding interface under the PCC wire [53].

**Figure 5 micromachines-14-01612-f005:**
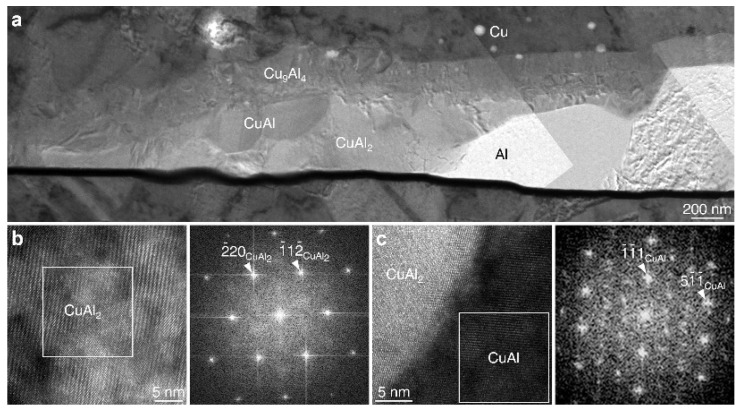
(**a**) BF TEM image of the fully consumed Al interface between the Cu-bonding wire and the Al pad after aging at 175 °C for 2000 h. HRTEM images of (**b**) CuAl_2_ and (**c**) CuAl grains, and FFT images of the area marked a white square in (**b**,**c**) [60].

**Figure 6 micromachines-14-01612-f006:**
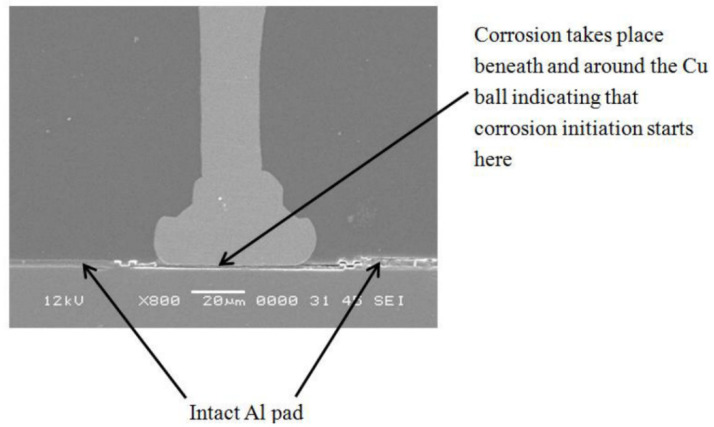
Cross-sectional analysis of Cu ball on Al pad revealing corrosion initiation beneath the Cu ball [70].

**Figure 7 micromachines-14-01612-f007:**
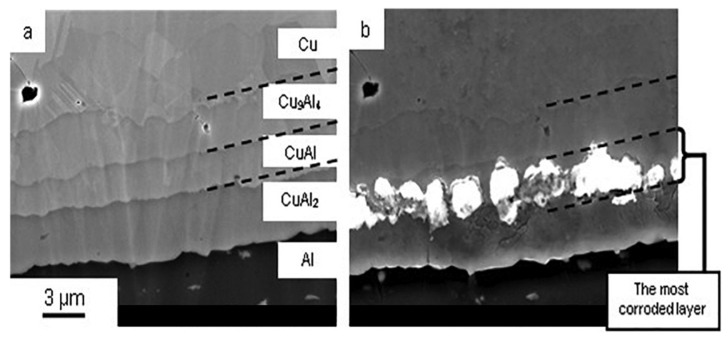
Cross-section of Cu/Al IMCs (**a**) before immersion in NaCl solution and (**b**) after immersion in NaCl solution [73].

**Figure 8 micromachines-14-01612-f008:**
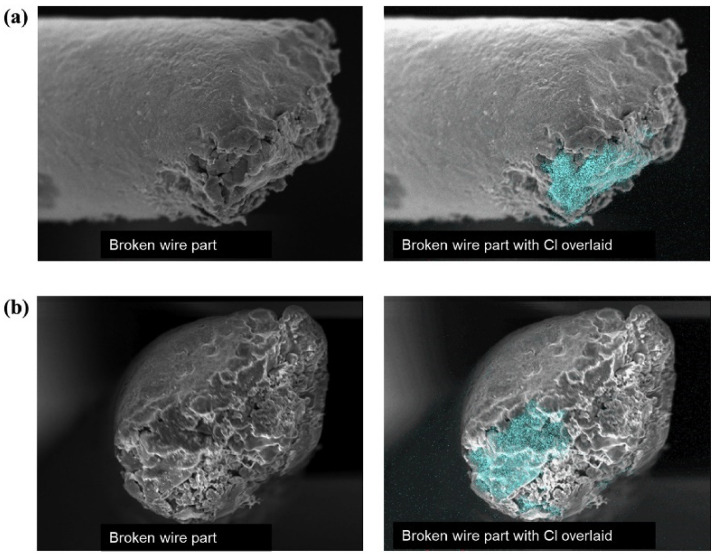
SEM/EDX mapping image of broken stitch Cu wire with chloride distribution. (**a**) Side view and (**b**) front view [74].

**Figure 9 micromachines-14-01612-f009:**
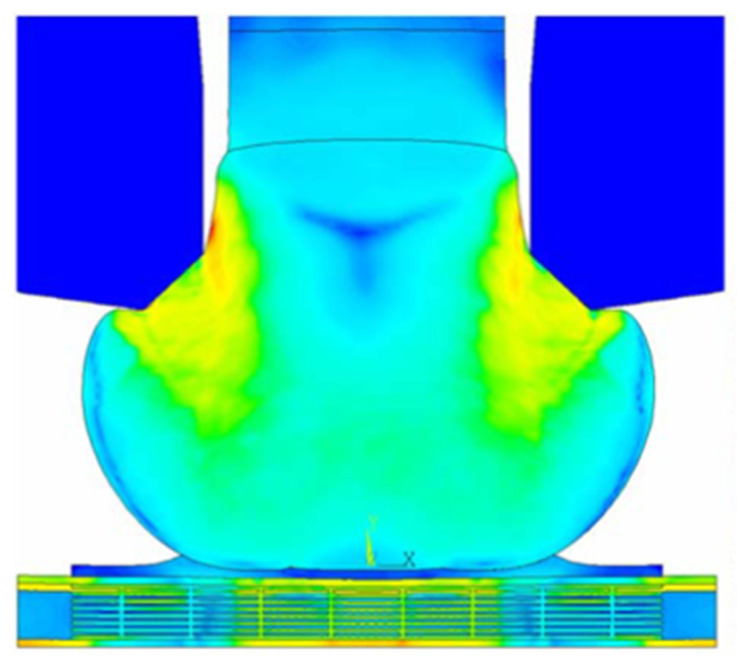
Overall stress distributions on the entire model (MPa) for EFO Cu wire. [82].

**Figure 10 micromachines-14-01612-f010:**
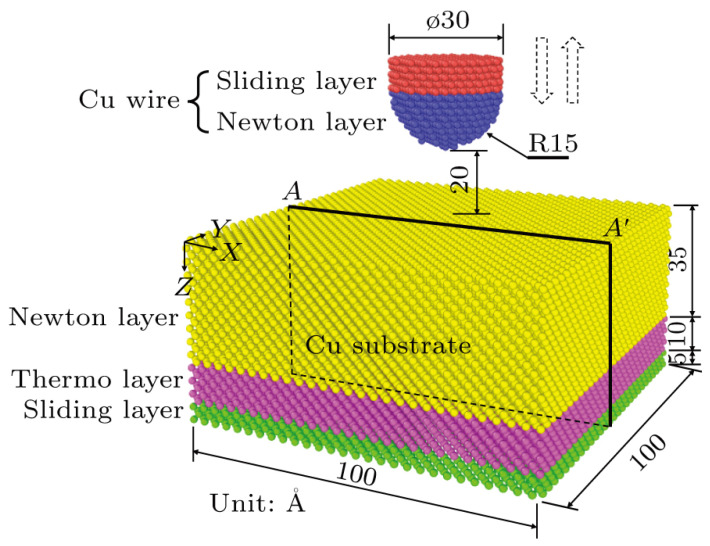
Geometries of the Cu–Cu wire bonding models [86].

**Table 1 micromachines-14-01612-t001:** Key performance of common bonding wire.

Material	Electrical Conductivity (Ω·m)	Thermal Conductivity (W/(m·K))	Tensile Strength (MPa)	Corrosion Resistance	Price
Cu wire	0.0017–0.0018	401–429	200–250	Moderate	Moderate
Al wire	0.028–0.036	205–230	70–400	Moderate	Low
Au wire	0.024–0.028	310–318	100–400	Good	High
Ag wire	0.015–0.020	419–429	100–400	Excellent	High

**Table 2 micromachines-14-01612-t002:** Comparison of basic performance between PCC wire and bare cu wire [21].

Cu Wire	Pd-Coated Cu	Bare Cu (No Coating)
Breaking load (mN)	112	107
Elongation (%)	11.8	12.1
Vickers hardness of wire	55	54
Electrical resistivity (10^−8^ Ω m)	1.9	1.9

## Data Availability

Not applicable.

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
