# Peer review of "Copper Wire Bonding: A Review"

_micromachines, 2023, doi:10.3390/mi14081612_

Round 1

Reviewer 1 Report

Minor changes ;

1:   Tables 1 and 2 should be at the center. More explanation for Table 2 is required. 

2:  Cu wire advantages should be in bullet points or numbers and presented in a proper way.

3:  Line 57 needs a reference and explanation.

4:  Please correct the shape and presentation of Figures 2, 3, and 4.

5:  Please correct line 211 reference.

6:  I would suggest that when you add an endurance limit or fatigue limit for Cu bonding It would have positive impacts on your work.

Major changes;

If organized including the electrical properties below, it will be a meaningful paper for circuit designers.

1. It is hoped that electrical analysis of bare cu, coated cu, insulated cu, and cu alloy for bonding applied to electronic packaging will be added.

2. It is hoped that each electrical analysis of the type of coated cu for bonding applied to electronic packaging will be added.

3. It is hoped that each electrical analysis of the type of insulated cu for bonding applied to electronic packaging will be added.

4. It is hoped that each electrical analysis of the type of cu alloy for bonding applied to electronic packaging will be added.

5. It is hoped that electrical analysis of broken stitch cu in Figures 5 and 6 will be added. And I hope to add an electrical characteristic comparison for each type of cu.

Author Response

Minor changes ;

(1) (2) (3) (4) (5) These have all been modified in the corresponding positions. Removes any confusion in the table.

(6) There are similar descriptions in Types of Cu bonding wire and bonding reliability.

Major changes;

(1) (2) (3) (4) Added at the end of pages 3, 4 and 5 of the paper.

(5) The types of copper wire used in both figures are included in the type of Cu-bonded wire and the electrical properties of the same type of copper wire are similar. Therefore no separate electrical characterisation has been carried out for these two types of wire.

Reviewer 2 Report

This manuscript provides an overview for copper wire bonding technology. It describes a variety of copper wire types, analyzes the reliability of copper wire bonding, and evaluates the effects of some factors on the reliability of copper wire bonding and the application of simulation to the reliability of copper wire bonding. I recommend to accepted this paper after a major revise addressing the following comments.

1. The graphical abstract would be valuable and important for the potential readers, especially in a review work. More abstract, summarizing images or tables would be included. This can make the paper more logical and give the reader a clearer understanding of the related works.

2. When the article introduces the various copper wire bonding technologies, it gives a brief introduction to “Bare Cu wire and Coated Cu wire”, but does not give an introduction to the “Insulated Cu wire and other technologies”, which is not very friendly to readers who are not familiar with the field.

3. All abbreviations (even well-known) should be explained at the first use.

4. The manuscript is not organized well enough and could be improved. In part 4.2 “Influence of Corrosion on the Reliability of Cu Wire Bonding”, there is only one subheading 4.2.1 “Factors Affecting the Corrosion Rate of Cu Wire” is described. It is utterly confusing that no other corresponding factor is listed.

5. In the page 5 “Application of Simulation in Cu Wire Bonding”, there is a formatting error in one of the paragraphs where there is no first line indentation. Also, there is no period in the last paragraph.

Author Response

(1)  Ok. Thank you for your advice. Added some representative figures

(2) Introduction to other copper wires has been added

(3)All right. Thank you for your valuable advice. Appropriate adjustments have been made.

(4)The corresponding changes have been added to pages 9 and 10.

(5)Thank you for your advice. Changes have been made.

Round 2

Reviewer 1 Report

All the comments meted well. If you present Table 3 on a single page, it makes your paper presentation better. 

Author Response

Relevant content has changed

Reviewer 2 Report

The manuscript is not organized well enough and could be improved. In part 4.2 “Influence of Corrosion on the Reliability of Cu Wire Bonding”, there is only one subheading 4.2.1 “Factors Affecting the Corrosion Rate of Cu Wire” is described. It is utterly confusing that no other corresponding factor is listed.

Author Response

The confusing parts of part4.2 have been modified, please check

Round 3

Reviewer 2 Report

No more comments.